# Telemental Health and Diverse Populations amid COVID-19

**Jiadong Yu \* and D. A. Bekerian**

California School of Professional Psychology, Alliant International University, Fresno, CA 93727, USA
\* Correspondence: jyu1@alliant.edu

**Definition:** Telemental health is defined as the delivery of psychological and mental health services via telecommunication technologies, including telephone-delivered therapy, videoconferencing, and internet-delivered programs. Research indicates that telemental health services are as effective as in-person services, and a dramatic increase in the use of telemental health has been observed during COVID-19. However, there are still persistent challenges and concerns about mental health providers' competencies, clients' data privacy, and legal and regulatory issues during this pandemic. Additionally, disparities in the use of telemental health services with diverse populations, based on factors such as age, gender, ethnicity, socioeconomic status, language, and culture, have been identified during this pandemic.

**Keywords:** telemental health; telemental health disparity; diverse population; COVID-19

## 1. Introduction

The Novel Coronavirus 2019 Disease (COVID-19) was an outbreak of severe acute respiratory syndrome (SARS), first reported in Wuhan, China [1]. The World Health Organization (WHO) declared this outbreak as a Public Health Emergency of International Concern (PHEIC) after COVID-19 infection rates rapidly spread across the world [2]. Initially, the public had to face this imminent health risk with limited information. In response, governments across the world decided to impose movement restrictions to curb COVID-19 spread [3]. The implementation of movement restrictions on daily life brought immediate challenges across all elements of society. In particular, the restrictions required mental health providers to offer and deliver care services via distance-based services rather than in-person services.

These distance-based services are collectively referred to as telemental health services. Telemental health refers to the use of telecommunication technologies to deliver mental healthcare services remotely, such as telephone-delivered therapy, videoconferencing, and internet-delivered programs. Services that can be delivered via telemental health include providing patients with health information and direct healthcare. This allows individuals to receive care from their own homes or other locations.

Telemental health has become an increasingly important tool in providing access to healthcare for the diverse population before and during the COVID-19 pandemic. However, disparities existed in access to telemental health services across different groups [4]. This entry aims to provide a general overview of the challenges and concerns that mental health providers are likely to face while delivering telemental health services and to examine disparities in telemental health services usage among diverse populations during COVID-19. However, the scope of the entry is limited to the existing literature that pertains to the use and delivery of telemental health services, and the low publication rate of Eastern countries may result in limited geographical diversity.

## 2. Telemental Health Services and COVID-19

### 2.1. Defining Telemental Health Services

Telemental health has a history dating back to the mid-20th century. Early forms of telemental health involved the use of telegraphs and radios to transmit medical information and provide a medical examination over distances to reduce unnecessary office visits [5]. As technologies progressed, new forms of mental health intervention delivery were discovered to improve the accessibility of mental healthcare.

Telephone, videoconferencing, and internet-delivered programs are now commonly utilized to provide mental health services [6]. Based on different types of communication, telemental health services can be categorized as synchronous and asynchronous [7]. Synchronous services refer to a mode of delivering health services where a real-time meeting occurs between the client and the health provider, such as telephone-based therapy and video-based therapy. Asynchronous services, in contrast, do not require a real-time meeting between the client and the service providers. Asynchronous services use the "store-and-forward" technique where the client's history report is sent to the mental health providers for diagnoses and treatment plans, which increase the flexibility for mental health providers to deliver the treatment [7,8]. For example, computerized or web-based assessments are the common methods of providing asynchronous telemental health services.

Telemental health services were mainly provided in rural and underserved areas to bridge the gap in mental health access in the United States [9]. Research demonstrated that Americans living in rural areas were eager to use the Internet for mental health services, especially where there were limited mental health providers available in their areas [10]. To this end, the Federal Communication Commission (FCC) in the United States commits millions of dollars each year to implement telemental health programs to help Americans keep connected to health services [11]. Several studies indicated increased satisfaction in receiving telemental health interventions across rural communities in the United States [12,13]. In Middle Eastern countries, telemental health services were increasingly used to meet the growing demands for mental healthcare, typically provided through hotlines and messaging services [14]. Telemental health was also found feasible in conflict-settings in several countries in the Eastern Mediterranean region by using low-cost technologies [15]. In Southeast Asia, telemental health also has the potential to achieve universal health coverage and has started to be implemented, such as in India, where the national telehealth network scheme has provided infrastructures for telemental health services in hospitals and remote healthcare facilities over the last two decades [16,17].

Telemental health has become increasingly important as a means of delivering mental health services [9,18]. For example, there were approximately twice as many U.S. mental health facilities offering telemental health in 2017 compared to 2010 [19]. Treatments delivered via telemental health have been found effective in addressing different mental health conditions, including substance abuse, depression, and anxiety [20–22]. Research suggests that the use of telecommunication results in higher completion rates of substance abuse treatment programs compared to in-person services; clients are also inclined to demonstrate more acceptance towards Internet-based treatment programs as it is more convenient and has more perceived confidentiality [23,24]. Multiple studies on telemental health have been conducted in different countries and demonstrated its effectiveness and efficacy, which further contributed to the increased need for telemental health services [25,26]. What is more important, as Western-developed countries have more access to care compared to developing countries, telemental health can bring evidence-based practices to the under-served and difficult-to-reach areas globally [27].

Along with the implementation of telemental health, studies have also found that telemental health services can reduce costs and overall evaluation time compared to in-person services [28,29]. For example, a longitudinal study found that the use of telemental health maintained the advantages in access and quality of mental health services with cost-efficiencies in a 2-year period [30]. Even though some studies would argue that the cost of telemental health services varied widely, there is good evidence that tele-

mental health increases access to mental health services and is as effective as in-person interventions [31,32].

It is evident that the use of telemental health services has progressed over the years. Telemental health has been proven effective in treating mental health conditions and has helped reach clients who had limited access to mental health services prior to the COVID-19 pandemic. Although the system to provide reliable access to effective telemental health services is still developing, the advent of COVID-19 inevitably accelerated the progress.

## 2.2. Telemental Health Competency Concern

The movement restrictions and social distancing guidelines brought immediate challenges for mental health providers and caused a rapid transition in the delivery of mental health services. In a survey conducted by the American Psychological Association between 23 April and 6 May in 2020, 76% of clinicians only provided remote services via phone, a designated telemental health platform, or videoconferencing software. In contrast, 3% of clinicians provided only in-person services [33]. Currently, many practicing mental health providers have reduced the time of seeing their clients in person and shifted to treating clients remotely because of the COVID-19 pandemic [34,35]. As there is an exponential increase in the usage of telemental health services, it is important to consider implementing telemental health training as some mental health providers might not be familiar with new technologies [36]. It is necessary for mental health providers to be comfortable with new technologies and be competent enough to utilize new technologies to facilitate individualized treatment [37].

Maheu et al.'s review (2019) presented the development of a measurable framework: the Coalition for Technology in Behavioral Science (CTiBS) for telemental health competencies based on literature reviews, technological advances, and clinical practice [38]. For example, mental health providers need to demonstrate their understanding of making decisions regarding the choice of using technology based on their own and their client's experience with technology. Maheu et al. also argued how crucial it is for mental health providers to understand how to utilize appropriate technology. Knowing how to use the technology helps maximize the therapeutic atmosphere during the meeting and spontaneously fosters a positive therapeutic relationship [38].

This framework identified the need for telemental health competency training; other studies have considered other factors that might impact new technology-based competencies, e.g., the length of time required for mental health providers to become familiar with telemental health technologies and workflows [39,40]. However, Edirippulige and Armfield's systemic review (2016) suggested there were limited programs related to the delivery of telemental health education and training. The authors found only nine studies that reported any formal telemental health training and education from universities, and public or private organizations [41]. The teaching and learning of telemental health competencies were not widely included in the formal education and training received by mental health professionals.

In response to the rapid shift during the COVID-19 pandemic, professional organizations such as American Telemedicine Association and American Psychological Association provided free online webinars, recorded content, and other resources as basic telemental health education to guide mental health providers through the transition [42,43]. A new training program was established and implemented at Virginia Commonwealth University to offer the training needed for telemental health, and the authors found the increased use of supervision via telephone or videoconferencing is a key factor for providing the necessary support for the trainees [44]. However, in India and Saudi Arabia standard practice protocols or guidelines need to be developed for mental health providers to ensure the quality of telemental health services [45,46].

Overall, the situation during the COVID-19 pandemic highlighted concerns regarding the paucity of telemental health training and education. The available literature suggests that with such a rapid transition there was limited time for mental health providers to

take educational training to develop telemental health competencies. There is also little evidence to show that short-term telemental health training programs have been adequately developed or studied for their effectiveness.

### 2.3. Privacy and Confidentiality Concern

With the pressure created by the COVID-19 pandemic, regulatory adjustments on the policy have been made by the United States government to meet the significantly increased needs for telemental health services [47]. For example, the United States Drug Enforcement Administration (DEA) has relaxed the standards for prescribing controlled substances when using telemental health services [48]. Although these adjustments can promote more access to mental health services, there are concerns about privacy and data confidentiality with the relaxation of standards.

A systemic review by Ftouni et al. (2022) suggested clients were concerned about their privacy and confidentiality regarding the use of telemental health services, which became another major barrier to telemental health visits [49]. For example, third-party commercial platforms such as Zoom, FaceTime, and Skype were commonly used for telemental health services. However, there are uncertainties about adequate encryption for these platforms for healthcare use [50]. Without adequate security and privacy protection, the mistrust from clients will impact the therapeutic relationship and impede the use of telemental health services [51]. The issues of clients' privacy and confidentiality must be studied in the future if telemental health is to be successful. With the increased use of third-party commercial platforms, technical issues became a significant barrier to accessing telemental health services.

For the purpose of promoting telemental health services during the COVID-19 pandemic, China established a trillion-dollar plan to expand digital infrastructure for stable connections, which includes developing 5G networks, industrial internet, and ultra-high-voltage (UHV) power transmission [52]. In Europe, the Netherlands, as one of the leading European countries in providing telemental health services, has been advancing the transition prior to COVID-19. In response to the pandemic, a COVID-19 online portal for sharing patient information was developed and implemented within months, and this platform further safeguards clients' privacy and was made available for free to all hospitals at launch [53]. However, in Sub-Saharan African countries, infrastructure and technical barriers, including electricity and internet, still exist during the COVID-19 pandemic [54]. Limited access to technology and poor internet connection were identified as important factors that prevented clients from engaging in telemental health services [55]. These factors can also impede the communication and interactions between the client and the mental health provider [49].

### 2.4. Legal and Regulatory Issues

In the United State, as telemental health services are often provided across state lines, mental health providers often needed to confront complex state licensure requirements [56]. With regulatory adjustments on policy, the Centers for Medicare and Medicaid Services (CMS) relaxed the restrictions for interstate telemental health services during COVID-19. However, some barriers still existed. For example, licensed professional mental health counselors were excluded from the exemptions [57]. Further, many states had not completely lifted restrictions to allow clinicians to practice across state lines; some states in the United States required health providers to submit an application for an emergency license [58]. The concerns many health providers have regarding licensure requirements across state boundaries further hinder the implementation of telemental health services.

In Middle Eastern countries, the healthcare system has been influenced by different legal and regulatory frameworks. The governmental structure in some countries was found to have a negative effect on the delivery of telemental health services, which further impacted clients' intention to use telemental health [59]. Moreover, regulatory and legislative guidelines are still in need of maneuvering the practice of telemental health

services to accommodate the rapid growth of telemental health during COVID-19 in these countries [60].

Reimbursement is one of the key components in considering the use of telemental health services. In the efforts to implement telemental health services, the Centers for Medicare and Medicaid Services expanded reimbursement for telemental health services during the COVID-19 pandemic [61]. Along with the recommendations from the CMS, some insurance companies started to expand coverage for telemental health services and telemental health services reimbursement [62]. Most health providers reported that they would continue to provide telemental health services if reimbursement continues [49]. However, future reimbursement for telemental health services still remains uncertain. Moreover, telemental health services were not reimbursed in all countries across the world. For example, there is no legislation for reimbursing telemental health services in countries such as Egypt and Brazil [63,64]. In Italy, a regulatory framework for telemental health started to develop along with the implementation of telemental health services in 2012 [65]. However, during COVID-19, similar to many Middle Eastern countries, telemental health services are still not considered essential levels of care in the public health system, causing the use of these services to primarily rely on out-of-pocket payment. Without support from government institutions, the implementation of telemental health services is likely to slow down in the future [60,66].

In summary, there are numerous challenges and concerns that need to be addressed for telemental health to continue to be an effective alternative to in-person treatment. Some of these challenges and concerns have been addressed in response to the COVID-19 pandemic, leading to the rapid shift to telemental health in certain countries. However, many of these challenges and concerns still persist. It is essential that all of these challenges and concerns are taken into consideration when implementing telemental health services in the future. Understanding these concerns and challenges will also aid in preparing mental health providers in the field of telemental health.

## 3. Disparities in the Use of Telemental Health Services

### 3.1. Age Disparities

The implementation of telemental health services is a promising solution to providing healthcare for people who have difficulties meeting health providers in person. Ideally, telemental health could help reduce health disparities in access to healthcare. However, disparities in the use of telemental health services during the COVID-19 pandemic have been found among diverse populations. Age, gender, race, socioeconomic status, and language factors impact on the equity of telemental health services across different populations [4,67–70].

For example, Jaffe et al.'s study found adults aged over 45 were less likely to use telemental health services compared to adults aged 18 to 44 years. This finding may be the result of older adults not being familiar with new technologies and resisting the use of telemental health services [67,71]. In addition, Luo et al.'s study (2021) suggested older adults were less likely to be prepared for telemental health services due to inexperience with technology [72,73]. Certainly, older adults preferred telemental health services to avoid the risks of in-person during the COVID-19 pandemic. However, the struggle with new technology impeded their attempt to use telemental health services [74]. Further studies are needed in the future to understand factors that contribute to the potential difficulties in using telemental health services among the older adults group.

### 3.2. Gender Disparities

It is evident that disparities in using telemental health services existed between older adults and young adults. However, there is also a significant gap in using telemental health services between male adults and female adults. For example, Lou et al. (2021) found more females preferred to use telemental health services [72]. One explanation for this disparity is that female adults worried more about health concerns during the COVID-19 pandemic,

while male adults were more concerned with the effects on the economy and society [75]. In contrast, Rahman et al.'s study (2022) in Bangladesh suggested the opposite finding. Their study suggested male adults more commonly used telemental health services compared to female adults. The authors argued the finding was the result of the potential gap in health awareness and literacy that existed between males and females in Bangladesh during the COVID-19 pandemic [76]. Nevertheless, these results indicated there is a disparity in using telemental health services between female adults and male adults that requires more attention.

### 3.3. Ethnic and Socioeconomic Disparities

Notable disparities in using telemental health services also exist in different ethnic groups and across different socioeconomic statuses. A study by Ramirez et al. (2020) suggested there was a significantly lower use of telemental health services among Hispanic clients compared to non-Hispanic White clients [77]. East and Southeast Asians also reportedly used telemental health services less often than non-Hispanic Whites [78]. In addition, African Americans represented a smaller proportion of those using telemental health services compared to non-Hispanic Whites [79]. These results indicated challenges in accessing and using telemental health services in minority communities during the COVID-19 pandemic. One of the possible explanations is the consequences of the digital divide.

The term digital divide refers to unequal access to technologies and uneven access to health information and communication among different groups [80]. There are two different levels regarding the digital divide. The first-level digital divide includes two different accesses: physical access and material access. Physical access, for example, is the access to broadband internet and smartphone devices, whereas material access refers to the ability to maintain access to the Internet over time [81]. When it comes to broadband internet, Black and Hispanic Americans reported less access compared to White Americans in Vogels's recent study (2021); a similar result was found regarding smartphone ownership, where Black and Hispanic Americans have less ownership compared to White Americans [82].

This first-level digital divide could be the potential reason for the gap in using telemental health services between different ethnic groups, where there is an evident disparity in accessing the Internet and the ownership of digital devices. Noticeably, low-income clients are likely to have similar difficulties in accessing the Internet and owning digital devices [83]. Despite the difficulty in physical access, the low-income group is also likely to struggle to maintain internet access in terms of material access. Studies indicated there is a disparity in using telemental health services among different income levels, where the low-income group was reportedly less likely to use telemental health services [84,85].

The second-level digital divide is defined as the ability to find reliable information on the Internet efficiently and effectively [86]. Additionally, it can be interpreted as a person's ability to determine the trustworthiness of information from online sources [80]. For example, Hispanics were less likely to seek internet health information compared to non-Hispanic Whites, although Hispanics agreed that health information from the Internet could be helpful [87]. The language barrier, one of the possible explanations, impeded non-English speakers' ability to find reliable information on the Internet. This language factor resulted in further disparity in using telemental health services.

### 3.4. Language Barrier

Language also has been identified as an important barrier to telemental health use in different ethnic groups, particularly in non-English-speaking clients [88]. For example, Asian Americans reportedly have more than 100 different languages and dialects across over 50 ethnicities [89]. There is a higher percentage of limited English proficiency found in Vietnamese, Chinese, and Korean clients [90]. The linguistic barrier could discourage Asian Americans and other minority groups with limited English proficiency from seeking telemental health services. The problem of a lack of bilingual services may account for the limited use of telemental health in these populations during the COVID-19 pandemic.

For example, unprofessional interpreters and missing bilingual services are the common causes of miscommunication between patients and health provider professionals, which further resulted in poor quality of care reported by linguistic minority groups [91]. The implementation of bilingual services has been found useful in improving the quality of healthcare and the level of satisfaction, which would further promote the use of telemental health services [92]. It is also important for policy makers and health providers to consider the implementation of bilingual services and promote linguistic equity when reducing the disparity in using telemental health services.

### 3.5. Cultural Stigma

Despite the consequences of the digital divide, another explanation for notable disparities that existed in different ethnic groups using telemental health services is the cultural stigma. A stigma can be defined as the implication of social disproval, which leads to discrimination and exclusion and affects an individual's self-esteem [93]. Stigmatization can be identified differently across cultures in diverse populations [94]. For example, seeking mental health services would be seen as an act of a "loss of face" in East Asian Culture, where "face" is known as a symbol of strong self-esteem [95]. What is more, Asian Americans reportedly had the lowest rate of seeking mental services compared to other ethnic groups [96] as they could be afraid of being culturally stigmatized. Therefore, seeking mental health services would incur discrimination among Asian Americans. Furthermore, there is a lack of awareness and acknowledgment of the need to seek mental health services in diverse populations [70]. For example, Asian Americans were concerned about burdening others when seeking mental health services, which led to an improper understanding of mental health [70]. Regardless of the advantages telemental health services have, it is impossible for the delivery of telemental health services to close the gap in access to healthcare under these circumstances. Future research is needed to focus on the different stigmas these populations are facing and consider cultural stigma when promoting equity in the use of telemental health services.

### 4. Conclusions and Prospects

The use of telemental health services has progressed over the years, and the effectiveness and efficacy of telemental health services have been proven before the COVID-19 pandemic. The COVID-19 pandemic forced an unprecedently rapid transition from in-person services to telemental health services. The transition has raised many questions.

Guiding organizations such as the American Counseling Association and the American Psychological Association have highlighted the importance of several factors for mental health providers to adopt telemental health services. For example, it is suggested that providers review the research on both efficacy and effectiveness to guide technology selection, be aware of the safety and crisis planning when using telemental health services, and be ready for the adaptations for interventions [97,98]. However, some important questions remain to be answered:

- Discussion must focus on the challenges and concerns (competency, privacy and confidentiality, legal issues, and future reimbursement) that mental health providers faced during the COVID-19 pandemic.
- Future research is needed for a reliable and valid system to address these concerns. More research will enable better solutions to the challenges and concerns that impede the implementation of telemental health services after COVID-19.
- Although the use of telemental health services can provide a solution to health disparities, the extant literature strongly argues that age, gender, race, socioeconomic status, language, and culture have prevented diverse populations from accessing and using telemental health services.
- A framework for telemental health equity incorporating individual, interpersonal, community, and societal variables is needed to address the disparities in using and accessing telemental health services among diverse populations.

We acknowledge that this entry is based on a non-systematic literature review. It may have selection bias when presenting these questions. Therefore, future research could minimize the bias by using a systematic or scoping review.

**Author Contributions:** Conceptualization, J.Y. and D.A.B.; writing—original draft preparation, J.Y.; writing—review and editing, J.Y. and D.A.B.; supervision, D.A.B. All authors have read and agreed to the published version of the manuscript.

**Funding:** This research received no external funding.

**Institutional Review Board Statement:** Not applicable.

**Informed Consent Statement:** Not applicable.

**Data Availability Statement:** Not applicable.

**Conflicts of Interest:** The authors declare no conflict of interest.

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
