# Peer review of "Telemental Health and Diverse Populations amid COVID-19"

_encyclopedia, doi:10.3390/encyclopedia3010017_

Round 1

Reviewer 1 Report

The article needs some improvements, the conclusion should highlight the theoretical and practical implications of the research results, the research limits. More I recommend create some recommendation of using telehealth services in future.

Reviewer 2 Report

This manuscript focuses on telehealth and specifically to it's applications during COVID-19 and issues affecting widespread adoption amongst diverse populations. Overall, this submission is a somewhat incomplete review of telehealth.

The "definition" section seems to end abruptly and incomplete. First, telehealth as described by the authors seems to be narrowly focused on tele-psychology or tele-mental health services. In general, telehealth is normally much more broadly defined to include other areas such as tele-nursing, tele-monitoring, tele-therapy, tele-radiology, etc. It would seem more is needed in the paper entry to cover these additional areas of telehealth. Also, this "abstract" section seems to miss summaries of the challenges for telehealth adoption and disparities across gender, ethnicity, etc. 

The authors describe 3 types of telehealth (ln 47)... synchronous, asynchronous, and remote monitoring. Traditionally speaking, remote monitoring can either be synchronous or asynchronous, but usually not thought of as a 3rd TYPE of telehealth. This reviewer suggests rewording, perhaps indicating that these are 2 types of telehealth "delivery methods." Starting in ln 57, it seems to suggest that telehealth was initiated to address mental health issues which is not quite correct as early applications were seen in tele-cardiology and tele-radiology. Mental health applications came later. Line 59 suggests telehealth started with the Internet which is also not true as early telehealth was conducted over telegraph, telephone, and satellites, decades before the Internet was widely available. The manuscript would benefit from more accurate historical presentation of the field. 

While the title seems to focus on the emergence of telehealth during early phases of COVID-19, there is surprisingly very little about applications of telehealth during this period (for home care, therapy, chronic illness management, etc). Line 87 suggests that in-person mental health has nearly stopped today. This seems to be an exaggeration and would benefit from some references. Yes, there is increased tele-pysch visits, but in-person visits have not stopped as suggested.  Further, much of the issues/challenges raised in section 2.2 (reimbursement, licensing, training, technology, etc) have been extensively discussed over the past 50 years of telehealth. In fact, many of the references cited are pre-COVID (2020) so it is not quite clear that the most currently thinking is incorporated in this manuscript entitled "telehealth" and "COVID-19". 

Overall, this entry is a somewhat incomplete review of telehealth and seems to miss an opportunity to summarize the tremendous growth in the field necessitated by COVID-19.

Reviewer 3 Report

This review paper covers the rapid expansion of telehealth services for mental health during the recent pandemic and associated challenges associated with transitioning to such telehealth mental health services.  Below are minor suggestions for strengthening the manuscript.

1.  Line 26-7. Change “Telehealth a term” to “Telehealth, a term”

2.  Line 34. Change “This entry paper aims to” to “The aims of this review are to”

3.  It would help to add subheading to some of the sections. Doing so will provide more clarity for the reader.

 For example, in Section 2.2. Challenges and Concerns, consider adding subheadings such as:

Line 87. Telehealth Competency

Line 116. Privacy and Confidentiality

Line 133. Technical Issues

Line 138. Regulatory issues (Licensure)

Line 147. Reimbursement

In Section 3, Disparities, consider adding subheadings such as:

Line 170. Age Disparities

Line 180. Gender Disparities

Line 192. Ethnic Disparities

Line 201. Digital Divide (Socioeconomic Disparities)

Line 243. Cultural Stigma

4. Line 170. Please double check the wording “aged 45-46” is the correct range intended. I suspect the author intended a wider older age range, starting with aged 45.

Reviewer 4 Report

This is a fairly well written and partly representative literature review of an important matter: telemental health services (partly but not exclusively in relation to the COVID-19 pandemic). Below find some comments for attention (in relation to order of appearance in the entry):

1.   The term telehealth is not specifically related to mental health services, as it also addresses other telehealth services such as telecardiology, teledermatology and more; hence, I recommend that it be replaced in the paper's title and elsewhere in the entry as appropriate with the term telemental health.

2. Methodologically, this entry may not require a systematic or scoping review, although that is preferable; if the entry is not revised to be a systematic or scoping review, at the very least it has to be clarified near the start of the entry and near its end that it is a selective literature review and what are the entry's limitations in relation to that, such as possible literature selection bias and more.  

3. A more comparative approach that is not focused primarily on the US is needed (recognizing that there is some of that in the entry, such as in relation to compensation), e.g., on pages 2 and 3 only the US is addressed but other countries have used telemental health services for decades now, such as Australia and Canada (who has had the largest user of telemental health services in the world for some time, i.e., the Ontario Telemedicine Network), which should be addressed and referenced.

4. It would be helpful to address for comparison and learning from that, even if briefly, non-mental health telehealth services, such as on page 4 in relation to compensation. 

5. I would be helpful to address efficiency and cost-effectiveness/benefit analyses and their findings in relation to telemental health services.

6. There are a few typos to correct.

Round 2

Reviewer 1 Report

-

Author Response

There are no comments or suggestions from the reviewer for the authors to address.

Reviewer 2 Report

This resubmission by the authors is much improved and tightly focuses on telemental health (vs. telehealth generally). The entry is a contribution to practitioners who may be seeking guidance on motivations and barriers to telehealth adoption. 

This reviewer only has minor editing suggestions:  "Rather" (vs "other"; line 28); capitalize "Health" in titles in section 2, 2.1, 2.2; "commits" (vs "committed"; ln 71); ln 88-89 starting with "For example..." is redundant with prior sentence; "telecommunication" (ln 90); "quality" (ln 102); "Lou et al" (ln 279); "physical access" vs "material access" explanation both refer to "access to broadband internet" and "access to internet" which seems to say the same thing; paragraph starting w/ln 319 talks about "reliable [health] information" but the rest of the paragraph talks about ethnic groups willingness to "seek internet health information" which are 2 different issues. 

Overall, this reviewer support publication of this manuscript. 

Author Response

Please see the attachment for the responses to reviewers 2 and 4.

Reviewer 3 Report

The authors have been responsive to the minor editorial suggestions.

Author Response

(The authors gave the same response as above.)
